# Placenta Accreta Spectrum Disorder Complicated with Endometriosis: Systematic Review and Meta-Analysis

**DOI:** 10.3390/biomedicines10020390

**Published:** 2022-02-06

**Authors:** Shinya Matsuzaki, Yutaka Ueda, Yoshikazu Nagase, Satoko Matsuzaki, Mamoru Kakuda, Sahori Kakuda, Hitomi Sakaguchi, Tsuyoshi Hisa, Shoji Kamiura

**Affiliations:** 1Department of Gynecology, Osaka International Cancer Institute, Osaka 541-8567, Japan; saho_y19@yahoo.co.jp (S.K.); hitomi.sakaguchi@oici.jp (H.S.); hisa-tu@mc.pref.osaka.jp (T.H.); kamiura-sh@oici.jp (S.K.); 2Department of Obstetrics and Gynecology, Osaka University Graduate School of Medicine, Osaka 565-0871, Japan; y.ueda@gyne.med.osaka-u.ac.jp (Y.U.); ynagase@gyne.med.osaka-u.ac.jp (Y.N.); mamorukakuda@gyne.med.osaka-u.ac.jp (M.K.); 3Department of Obstetrics and Gynecology, Osaka General Medical Center, Osaka 558-8558, Japan; satoko_tsuru@yahoo.co.jp

**Keywords:** placenta accreta spectrum, endometriosis, pelvic adhesion, assisted reproductive technology, systematic review

## Abstract

This study aimed to assess the relationship between placenta accreta spectrum disorder (PASD) and endometriosis. The relationships among pregnancy, assisted reproductive technology (ART), placenta previa, ART-conceived pregnancy and PASD were also determined. A systematic literature review was conducted using multiple computerized databases. Forty-eight studies (1990–2021) met the inclusion criteria. According to the adjusted pooled analysis (*n* = 3), endometriosis was associated with an increased prevalence of PASD (adjusted odds ratio [OR] 3.39, 95% confidence interval [CI] 1.96–5.87). In the included studies, the ART rate ranged from 18.2% to 37.2% for women with endometriosis. According to the adjusted pooled analysis, women who used ART were more likely to have placenta previa (*n* = 13: adjusted OR 2.96, 95%CI, 2.43–3.60) and PASD (*n* = 4: adjusted OR 3.54, 95%CI 1.86–6.76) than those who did not use ART. According to the sensitivity analysis using an unadjusted analysis accounting for the type of ART, frozen embryo transfer (ET) was associated with an increased risk of PASD (*n* = 4: OR 2.79, 95%CI, 1.22−6.37) compared to fresh ET. Endometriosis may be associated with an increased rate of PASD. Women with placenta previa complicated with endometriosis who conceived using frozen ET may be a high risk for PASD.

## 1. Introduction

For women with placenta accreta spectrum disorder (PASD), massive hemorrhage is a severe complication, with intraoperative blood loss possibly exceeding several liters and leading to disseminated intravascular coagulation and, ultimately, maternal death [1,2,3,4]. There is also a high risk of bladder injury and ureter injury; therefore, cesarean hysterectomy without any attempt to remove the placenta is often required to avoid these complications [2,5,6,7]. Endometriosis is a frequent benign gynecologic disease wherein endometrial-like glands and stroma grow outside the uterus [8,9].

The estimated incidence of endometriosis in women of reproductive age is around 5–15%, and endometriosis may be increasing in pregnant women [10,11,12,13]. Moreover, live birth rates have improved with assisted reproductive technology (ART) and previous studies have shown that 25–50% of infertile women suffer endometriosis [10,11,12,13,14]. 

Pregnant women with endometriosis are more likely to use ART to conceive than those without endometriosis; however, ART-conceived pregnancies are associated with an increased prevalence of PASD [15]. Furthermore, recent systematic reviews have reported that women with endometriosis are more likely to have placenta previa than those without endometriosis. Placenta previa involves a high risk of postpartum hemorrhage, and approximately half of all placenta previa cases result in postpartum hemorrhage [16]. Additionally, placenta previa is associated with an increased rate of PASD and is the most significant risk factor for PASD [17]. Therefore, it is possible that women with endometriosis have a higher prevalence of PASD than those without endometriosis [14,18,19]. Previous studies have shown that women with endometriosis who underwent gynecologic surgery have had increased rates of ureteral injury and prolonged operative times compared to those without endometriosis [20,21].

As shown in Appendix A [22,23,24,25,26], women with endometriosis who have PASD often have extrauterine posterior adhesions, thus making it difficult to exteriorize the uterus. Based on our experience, women with PASD and endometriosis have poor surgical outcomes because of the difficulty of cesarean hysterectomy. Therefore, we believe that endometriosis may be correlated with adverse surgical outcomes of women with placenta previa and PASD. Recently, endometriosis has been consistently found to be associated with an increased risk of placenta previa [27,28]. Although the association between endometriosis and an increased risk of placenta previa is robust [22,23,24,25,26], the association between PASD and endometriosis is unclear. Moreover, the surgical outcomes of women with PASD complicated with endometriosis are unclear.

We performed a systematic review of computerized databases from their inception to 31 October 2021. This study aimed to assess the effect of endometriosis on the incidence of PASD. Since women with endometriosis often conceive using ART, we also determined the effect of ART on the prevalence of placenta previa and PASD. If no studies regarding the outcome of interest were identified during our systematic review, then a narrative review or our opinion about the topic regarding pregnant women with endometriosis was used.

## 2. Materials and Methods

Since endometriotic lesions are difficult to diagnose in pregnant women, the current study identified endometriosis in pregnant women with the following criteria [29]: (i) histologically confirmed endometriosis; (ii) endometriosis diagnosed by clinical findings (e.g., ultrasound, pelvic adhesions, posterior uterine wall adhesions, or the existence of an endometriotic cyst); (iii) endometriosis suspected clinically or histopathologically during cesarean delivery; and (iv) women identified with endometriosis by using the International Classification of Diseases code. Severe endometriosis was defined as deep infiltrating endometriosis or revised American Society for Reproductive Medicine stage III or IV endometriosis.

### 2.1. Systematic Literature Review

A systematic literature search was performed to examine the effect of endometriosis on the incidence of PASD. The same search terms used for endometriosis during our previous systematic review were used during this study (Appendix A) [29]. The outcomes of interest during this study were as follows: (i) the effect of endometriosis on the prevalence of PASD; (ii) the effect of endometriosis on surgical outcomes of patients with placenta previa and endometriosis; (iii) the effect of ART on the incidence of placenta previa; (iv) the influence of ART on the PASD rate.

In compliance with the 2020 edition of the Preferred Reporting Items for Systematic Reviews and Meta-Analyses statement [30], a systematic literature search was performed using PubMed, the Cochrane Central Register of Controlled Trials (CENTRAL), and Scopus from their inception to 31 October 2021, using Medical Subject Headings (MeSH terms) (if applicable) and words related to placenta previa complicated by endometriosis. This systematic review was not pre-registered.

### 2.2. Eligibility Criteria, Information Sources, and Search Strategy

Studies were screened by checking the titles and abstracts of relevant articles, as previously described [31,32,33]. All abstracts were screened by the authors (Sh.M. and Sa.M.) using the the MeSH terms and the PubMed and Cochrane databases to identify studies that examined the following associations: endometriosis and PASD; ART and placenta previa; ART and PASD (Appendix A).

### 2.3. Study Selection

Studies were included if they met the following criteria: (i) comparative study was performed that examined the outcomes of interest between an experimental group and a control group (e.g., comparison of women with and without endometriosis, ART *versus* spontaneous conception, frozen embryo transfer [ET] *versus* fresh ET, etc.); (ii) endometriosis as defined according to this study; (iii) pelvic adhesions caused by suspected endometriosis (most patients had a histopathological or clinical diagnosis); (iv) the influence of endometriosis on the prevalence of PASD was determined; (v) surgical outcomes of women with endometriosis complicated by PASD were discussed; (vi) the association between ART and placenta previa was examined; and (vii) the effect of ART on the frequency of PASD was determined.

The exclusion criteria were as follows: (i) insufficient information to clearly identify the outcomes of interest; (ii) the definition of endometriosis was unclear or did not meet the definition used by this study; (iii) patients’ background were restricted (e.g., cases were restricted to women with prior cesarean delivery, or polycystic ovary syndrome, or oocyte donation, etc.); (iv) non-English articles; (v) and case reports, conference abstracts, case series, reviews, systematic reviews, and meta-analyses.

### 2.4. Data Extraction

All data were extracted by the author (Sh.M.). The year of the study, first author’s name, study location, number of included cases, definition of endometriosis, and outcomes of interest were recorded. The data to be included in the analysis were verified by another author (Y.N.).

### 2.5. Analysis of Outcome Measures and Assessment of Bias Risk

The primary aims of this review were to focus on the clinical research surrounding PASD complicated with endometriosis and to focus on the studies assessing the effect of ART on the prevalence rates of placenta previa and PASD. The primary outcome was the surgical outcomes of women with endometriosis complicated by PASD. The secondary aims of this study were as follows: (i) to determine the estimated prevalence of PASD for patients with endometriosis; (ii) to determine the effect of endometriosis on the diagnosis of PASD; (iii) and to determine the proposed surgical technique for endometriosis patients with PASD during cesarean delivery. 

Risk of bias assessment was performed using the Risk Of Bias In Non-randomized Studies-of Interventions tool (ROBINS-I) as previously described [34,35,36].

### 2.6. Meta-Analysis 

Hazard ratios by using 95% confidence intervals (CIs) and odds ratios (ORs) were calculated with the outcomes of interest. Study heterogeneity was analyzed using the *I*^2^ statistic to determine the total deviation percentage among studies. In accordance with the *Cochrane Handbook for Systematic Reviews of Interventions* (version 6.0), heterogeneity was determined based on the *I*^2^ value as: low heterogeneity (*I*^2^: 0−30%); moderate heterogeneity (*I*^2^: 30−60%); substantial heterogeneity (*I*^2^: 50−90%); and considerable heterogeneity (*I*^2^: 75−100%) [37].

A meta-analysis was also performed. All graphics were constructed using RevMan version 5.4.1 software (Cochrane Collaboration, Copenhagen, Denmark). During the pooled analysis, a fixed-effects model was used for low heterogeneity, and a random-effects model was used for moderate to considerable heterogeneity.

### 2.7. Statistical Analysis

The chi-squared test or Fisher’s exact test was used to analyze differences in patients’ characteristics between the experimental and control groups. All statistical analyses were based on two-sided hypotheses. A *p*-value < 0.05 was considered statistically significant [38]. The SPSS version 28.0 (IBM Corp., Armonk, NY, USA) was used in the analyses.

## 3. Results

Figure 1 illustrates the study selection scheme. Overall, 1175 studies were examined. Four studies were excluded due to the presence of overlapping cases [39,40,41,42]. Forty-eight studies that comprised 34,603,164 pregnancies and 59,241 pregnancies with endometriosis met the inclusion criteria for the descriptive analysis [26,27,43,44,45,46,47,48,49,50,51,52,53,54,55,56,57,58,59,60,61,62,63,64,65,66,67,68,69,70,71,72,73,74,75,76,77,78,79,80,81,82,83,84,85,86,87,88]. 

### 3.1. Study Characteristics

Appendix A summarizes the metadata of the 48 included studies [26,27,43,44,45,46,47,48,49,50,51,52,53,54,55,56,57,58,59,60,61,62,63,64,65,66,67,68,69,70,71,72,73,74,75,76,77,78,79,80,81,82,83,84,85,86,87,88] and 4 excluded studies [39,40,41,42]. The included studies were published between 1990 and 2021, and all studies were retrospective (*n =* 48). Fourteen of 48 studies were nationwide studies, and no study was a randomized, controlled study. Approximately one-third of the studies were performed in Europe (*n* = 17; 35.4%) [27,46,48,53,54,57,60,62,64,70,71,72,73,77,79,83,86], followed by Japan (*n* = 11; 22.9%) [26,52,55,56,58,59,68,74,81,82,88], China (*n* = 10; 20.8%) [43,45,47,49,61,66,67,69,78,80], the United States (*n* = 4; 8.3%) [50,51,75,87], Israel (*n* = 2; 4.2%) [63,84], Australia (*n* = 2; 4.2%) [76,85], Canada (*n* = 1; 2.1%) [65], and Iran (*n* = 1; 2.1%) [44].

#### 3.1.1. Risk of Bias of Included Studies

The risk of bias assessment for the comparative studies is shown in Appendix A. Of those (*n* = 48), a possible moderate publication bias (moderate quality) in 36 studies and severe publication bias (low quality) in the other 12 studies were observed.

#### 3.1.2. Number of Studies: Primary Outcome

A systematic literature search was performed to identify studies that included the outcomes of interest. There were four regarding the effect of endometriosis on the prevalence of PASD. There were zero studies regarding the effect of endometriosis on the surgical outcomes of patients with PASD and endometriosis. There were 27 studies regarding the effect of ART on the rate of placenta previa. There were 15 regarding the influence of ART on the incidence of PASD.

#### 3.1.3. Number of Studies: Secondary Outcome

There was one study regarding the estimated prevalence of PASD for patients with endometriosis. There were zero studies regarding the effect of endometriosis on the diagnosis of PASD. There were zero studies regarding the proposed surgical technique for endometriosis patients with PASD during cesarean hysterectomy. Only one study regarding the secondary outcomes of this study was identified.

### 3.2. Epidemiology and Outcomes

#### 3.2.1. Prevalence of PASD with Endometriosis

We performed a systematic literature search of the epidemiology of patients with PASD and endometriosis. During this analysis, only nationwide studies were included. To date, only one study performed in Europe has examined the prevalence of PASD for women with endometriosis during pregnancy [27]. A study by Berlac indicated that 7 of 73,272 (0.01%) women had endometriosis with PASD [27]. To discuss this topic, we added a narrative review of the prevalence of PASD for the general population. 

#### 3.2.2. Population-Based Prevalence of PASD for the General Population

Recently, several studies have examined the prevalence of PASD for the general population (Table 1) [17,56,89,90,91,92].The estimated overall prevalence of PASD reported by these studies varied from 0.05% to 0.84%. Nevertheless, some nationwide studies support the notion that the prevalence of PASD is increasing [17,91]. Regarding the prevalence of endometriosis during pregnancy, two nationwide studies have shown rates of 0.9% [48] and 1.8% [27]. These results suggest that PASD complicated with endometriosis is rare.

Although we believe that these studies are useful for estimating the prevalence of PASD for women with endometriosis, it should be noted that the accuracy of the diagnosis of endometriosis with PASD is unclear in nationwide studies (all studies identified eligible patients using diagnostic codes or clinical diagnosis). Therefore, future studies examining the nationwide prevalence of PASD for women with endometriosis with an accurate diagnosis are warranted.

### 3.3. Primary Outcome: Association between Endometriosis and PASD

We identified four comparator studies with moderate quality that examined the effect of endometriosis on the prevalence of PASD (Table 2) [27,47,63,64]. Of those four studies, one was a population-based study and three were retrospective studies. Furthermore, among those four studies, three clarified the definition of endometriosis and one clarified the definition of PASD. According to the unadjusted, pooled, random-effects analysis, endometriosis was associated with an increased rate of PASD (Figure 2A) (*n* = 4, OR 3.97, 95%CI 1.30–12.11; heterogeneity: *p* = 0.02, *I*^2^ = 70%). According to the adjusted, pooled, fixed-effects analysis, endometriosis was associated with an increased prevalence of PASD (Figure 2B) (*n* = 3, OR 3.39, 95%CI 1.96–5.87; heterogeneity: *p* = 0.34, *I*^2^ = 8%).

### 3.4. Primary Outcome: Association between ART and Placenta Previa

A meta-analysis using 27 comparative retrospective studies (7 of low and 20 of moderate quality) was conducted to determine the influence of ART on the prevalence of placenta previa (Appendix A, Table 3) [26,43,47,48,49,51,52,53,54,55,57,60,61,62,65,66,67,69,72,73,78,79,82,85,86,87,88]. Since considerable heterogeneity of studies was observed, a random-effects analysis was used. According to the unadjusted pooled analysis (*n* = 27), women who conceived using ART were more likely to have placenta previa than those who conceived without ART (Figure 3A) (OR 3.47, 95%CI 2.74–4.39; heterogeneity: *p* < 0.01, *I*^2^ = 99%). 

During the adjusted pooled analysis, we performed a random-effects analysis for considerable heterogeneity. The results of this study were similar to those of the unadjusted analysis (Figure 3B) (OR 2.96, 95%CI 2.43–3.60; heterogeneity: *p* < 0.01, *I*^2^ = 92%) of women with endometriosis and women without endometriosis [26,27,48,63,66,68,70,81,85,93,94,95]. Based on the results of these studies, ART-conceived pregnancy was associated with a significantly increased rate of placenta previa.

### 3.5. Primary Outcome: Association between ART and Placenta Previa (Sensitivity Analysis)

To enhance the robustness of the results that showed a positive association between ART and placenta previa, we conducted a sensitivity analysis comparing the following groups: (i) ART with endometriosis and ART without endometriosis; (ii) women with endometriosis who conceived using ART and women with endometriosis who conceived spontaneously; (iii) those who used frozen ET and those who used fresh ET; and (iv) the hormone replacement cycle (HRC) and the normal cycle (NC). Among these analyses, we considered that the comparison of women with endometriosis who used ART and women without endometriosis who used ART was essential to evaluating the effect of ART on women with endometriosis.

#### 3.5.1. Effect of Endometriosis on the Prevalence of Placenta Previa for Women Who Conceived Using ART

Seven retrospective studies were included to determine the effect of endometriosis on the prevalence of placenta previa by comparing women with endometriosis who conceived using ART (*n* = 1914) and women without endometriosis who conceived using ART (*n* = 7917) (Table 4) [47,68,70,71,81,83,85]. According to the unadjusted random-effects analysis, endometriosis was associated with a higher rate of placenta previa (Figure 4A) (*n* = 7: OR 4.09, 95%CI 1.93–8.69; heterogeneity: *p* < 0.01, *I*^2^ = 73%). These results suggest that endometriosis has an additive effect that increased the rates of placenta previa for women who conceived using ART.

#### 3.5.2. Effect of ART on the Prevalence of Placenta Previa for Women with Endometriosis Who Conceived Using ART

One population-based study was eligible for use to examine the effect of ART on the prevalence of placenta previa for women with endometriosis who conceived using ART (*n* = 6934) and women with endometriosis who conceived spontaneously (*n* = 31,101) [48]. Women with endometriosis who conceived using ART were more likely to have placenta previa than those who conceived spontaneously (Figure 4B) (*n* = 1: OR 2.94, 95%CI 2.55–3.38). Although this result was based on only one study, it is possible that ART has an additive effect that increased the frequency of placenta previa for women with endometriosis.

#### 3.5.3. Rates of Placenta Previa according to the Type of ART

For a more in-depth examination of the effect of ART, a sensitivity analysis was conducted according to the type of ART (Table 4 and Appendix A; fresh ET *versus* frozen ET). During this analysis, 10 studies investigating 104,739 pregnancies conceived using frozen ET and 83,663 pregnancies conceived using fresh ET were identified [44,52,58,72,74,75,76,77,80,85]. For this analysis, an unadjusted random-effects analysis was conducted. Women who conceived using frozen ET were less likely to have placenta previa (Figure 4C) (*n* = 10: OR 0.71, 95%CI 0.57–0.88; heterogeneity: *p* = 0.03, *I*^2^ = 52%) than women who conceived using fresh ET. 

Four studies determined and compared the rates of placenta previa for women who conceived using frozen ET with the HRC and for women who conceived using frozen ET with the NC. This comparison showed a similar rate of placenta previa (Figure 4D) (*n* = 4: OR 1.14, 95%CI 0.72–1.80; heterogeneity: *p* = 0.02, *I*^2^ = 54%) for the two groups.

### 3.6. Association between ART and PASD

During our previous systematic review, we examined the effect of ART on the incidence of PASD [15]. However, because we did not collect data regarding endometriosis or examine the influence of endometriosis on the rate of PASD during our previous study [15], we performed a meta-analysis by revising the keywords for the literature search to include women with endometriosis, updating the study period, and correcting the data regarding endometriosis.

Of the included studies, only two clarified the rate of endometriosis. These limited data are inadequate for determining the effect of endometriosis on women who used ART (Appendix A). As shown in Table 5, 15 studies (4 of low and 11 of moderate quality) examined the effect of ART on the incidence of PASD for pregnant women [43,47,49,50,51,55,56,59,69,82,84]. Of those 15 studies, 10 compared the PASD rates of women who used ART and those of women who did not use ART, and five determined the prevalence of PASD according to the type of ART (four compared frozen ET and fresh ET and one compared HRC and NC). 

#### 3.6.1. ART-Conceived Pregnancy *versus* Pregnancy Conceived without ART

Using the 10 studies that compared ART-conceived pregnancy (*n* = 284,520) and pregnancy without the use of ART (*n* = 16,808,902), we performed an unadjusted random-effects analysis to examine the effect of ART on the prevalence of PASD (Figure 5A). According to those 10 studies, ART-conceived pregnancy was associated with an increased rate of PASD (*n* = 10: OR 4.22, 95%CI 2.95–6.03; heterogeneity: *p* < 0.01, *I*^2^ = 92%). The results of the adjusted random-effects analysis were similar to those of the unadjusted analysis (Figure 5B) (*n* = 4: OR 3.54, 95%CI 1.86–6.76; heterogeneity: *p* < 0.01, *I*^2^ = 91%).

#### 3.6.2. Sensitivity Analysis of Frozen ET *versus* Fresh ET and of HRC *versus* NC

During the sensitivity analyses, the effect of ART on the PASD rates was examined according to the ART type. The comparison of frozen ET and fresh ET showed that frozen ET was associated with a higher incidence of PASD (Figure 6A) (*n* = 4: OR 2.79, 95%CI, 1.22–6.37; heterogeneity: *p* < 0.01, *I*^2^ = 79%). 

The comparison of the HRC and NC showed that HRC was associated with an increased rate of PASD (Figure 6B) (*n* = 1: OR 5.76, 95%CI, 3.12–10.64). These results suggest that frozen ET with the HRC may be the most significant type of ART that is associated with higher rates of PASD.

### 3.7. Secondary Outcomes: Effect of Endometriosis on the Diagnosis of PASD

#### 3.7.1. Systematic Review Results

We performed a systematic literature search to find a study that examined the effect of endometriosis on the diagnosis of PASD. However, we could not find any study that has examined the effect of endometriosis on the accuracy of the PASD diagnosis.

#### 3.7.2. Narrative Review of the Diagnosis of PASD for Women with Endometriosis

To date, the effect of endometriosis on the PASD diagnosis remains unclear. Since approximately 20% to 30% of women with endometriosis conceived using ART, we reviewed the effect of ART on the accuracy of the PASD diagnosis. According to a retrospective study performed in Japan, the diagnostic accuracy for PASD after ART with placenta previa was examined using magnetic resonance imaging (MRI) [19]. During this study, the antenatal diagnosis rate using MRI for women with PASD who conceived using ART was significantly lower than that for women with PASD who conceived without ART (2/9 [22.2%] *versus* 18/19 [94.7%]; *p* < 0.01) [19]. 

Another study performed in the United States that investigated the antenatal diagnosis rate of PASD using ultrasonography [50] showed that the antenatal diagnosis of PASD after ART was significantly lower than that for women with PASD who conceived without ART (<4/31 [<12.9%] *versus* 38/81 [46.9%]; *p* < 0.01) [50]. These results suggest that ART-induced PASD has the potential to be associated with lower diagnostic accuracy compared to PASD without ART.

## 4. Discussion

### 4.1. Principal Findings

This study resulted in four principal findings. First, endometriosis is associated with an increased incidence of PASD; Second, ART, especially fresh ET, is associated with an increased rate of placenta previa; Third, ART, especially frozen ET during the HRC, is associated with a higher prevalence of PASD; Fourth, the surgical outcomes of cesarean hysterectomy for women with PASD and endometriosis have not been reported. Although the relationship between endometriosis and the increased incidence of PASD is unique, the mechanism of the increased rate of PASD is unresolved, and a meta-analysis evaluating the effect of endometriosis and excluding confounding risk factors for PASD (advanced maternal age, increased rate of ART, increased rate of placenta previa) was impossible. Therefore, future studies are warranted.

### 4.2. Strengths and Limitations

This study is likely the first to focus on the effect of endometriosis on the prevalence of PASD. This study revealed that endometriosis is correlated with a higher incidence of PASD. We evaluated the association between ART and placenta previa as well as the association between ART and PASD. Because approximately 20% to 30% of women with endometriosis conceive using ART, these investigations are useful for clinicians. Notably, no meta-analysis of the effect of ART on the frequency rates of placenta previa and PASD and no sensitivity analysis of endometriosis have been reported previously. Therefore, our results may be useful for clinicians.

This study had several notable limitations. First, because all eligible studies were retrospective, unmeasured bias may exist. Other possible confounding factors in the eligible studies were the definitions of endometriosis that varied among studies, the limited number of studies that examined the association between endometriosis and PASD, the fact that most meta-analyses had considerable heterogeneity among the studies, and the low quality of the diagnoses of PASD and endometriosis in some of the studies. These factors may have created severe bias in the study and in the considerable heterogeneity among the studies. 

Second, although the surgical outcomes of women with PASD complicated with endometriosis during cesarean hysterectomy are essential to the impact of endometriosis on the rate of PASD, this information was lacking due to the absence of previous studies. Future studies that examine the surgical outcomes of PASD complicated with endometriosis are warranted. Third, because no studies have examined the effect of endometriosis on the rate of PASD according to the severity of endometriosis, the association between severe endometriosis and PASD and the association between non-severe endometriosis and PASD are still unclear. Further studies are necessary to identify these associations. 

Fourth, we could not examine the effect of endometriosis on the incidence of PASD after excluding confounding factors. Because women with endometriosis are more likely to use ART to conceive, more likely to be of advanced maternal age, and more likely to have placenta previa compared to those without endometriosis [14,96], it is essential to exclude the cofounding factors. However, none of the studies performed a multivariate analysis with adjustments for the obstetric background; therefore, our analysis cannot characterize endometriosis as a risk factor for PASD after excluding the confounding factors [27,47,63,64].

Finally, during the analysis of ART and PASD, although various obstetric factors (advanced maternal age, placenta previa, prior uterine surgery, etc.) are associated with an increased rate of PASD, none of the studies performed a multivariate analysis with adjustments for these obstetric factors; therefore, our analysis cannot illustrate ART as a risk factor for PASD after excluding the confounding factors [43,47,49,50,51,55,56,59,69,82,84]. These are notable limitations that others should be aware of when interpreting the results of this study.

### 4.3. Comparisons with the Existing Literature

The association between endometriosis and an increased risk of placenta previa is robust [23,24,29,97,98]. Notably, placenta previa is the most significant risk factor for PASD [15,17,98,99,100,101]. Although our study identified that endometriosis is a risk factor for PASD, no studies have examined the influence of endometriosis on the rate of PASD after excluding the effect of the increased rate of placenta previa. Further studies that adjust for the effect of the confounding factors for PASD (placenta previa, ART-conceived pregnancy, prior uterine surgery, etc.) are warranted to examine the association between endometriosis and PASD.

#### 4.3.1. Association between Endometriosis and ART

Endometriosis has been reported to affect approximately 10% to 15% of women of reproductive age [102]. According to our included studies, the rate of ART-conceived pregnancy for women with endometriosis ranged from 18.2% to 37.2% [26,53,66]. These rates are similar to those reported by previous studies (11.9–26.0%) [103,104,105]. Although the definition of endometriosis is different among studies, approximately one-fourth of women with endometriosis conceive using ART [26,53,66,102,103,104,105]. Therefore, we consider the relationship between ART and placenta previa, as well as that between ART and PASD, essential to examining the influence of ART on the frequency of PASD.

#### 4.3.2. Association between ART and Placenta Previa

Some systematic reviews have examined the association between ART and placenta previa. ART-conceived pregnancy was associated with an increased rate of placenta previa compared to pregnancies conceived without ART [97,106]. Although the rate of placenta previa was similar for frozen ET and fresh ET (OR 0.70, 95%CI, 0.46–1.08) according to a previous systematic review in 2018, our study showed that frozen ET is associated with a decreased rate of placenta previa (OR 0.71, 95%CI, 0.57–0.88) compared to fresh ET [97]. Since our systematic review is the latest review of this topic, the inconsistency in the results may be attributable to the difference in the number of included studies. 

A previous retrospective study that determined an association between endometrial thickness and placenta previa found that women with endometrial thickness more than 12 mm were more likely to have placenta previa (adjusted OR 3.74, 95%CI, 1.90–7.34) than women with endometrial thickness less than 9 mm [76]. Previous studies have reported that the endometrial thickness was thinner in women who conceived with frozen ET than in women who conceived with fresh ET [76,107,108]. Therefore, a possible mechanism for the decreased rate of placenta previa for women who conceived using frozen ET is the thinner endometrial thickness. 

#### 4.3.3. Association between ART and PASD

During our previous systematic review, we examined the effect of ART-conceived pregnancy on the prevalence of PASD compared with the effect of pregnancy conceived without ART [15]. During this updated systematic review, we attempted to examine the effect of endometriosis on the incidence of PASD for women with ART, which was impossible due to the lack of studies. Although we added three new studies to this analysis, the results were similar to those of our previous study [15]. Further studies are necessary to examine the effect of endometriosis on the prevalence of PASD with ART-conceived pregnancy.

#### 4.3.4. Possible Mechanism of the Increased Rate of PASD for Women with Endometriosis

As shown in Figure 7, the results of our study revealed a possible mechanism for the increased rate of PASD for women with endometriosis. We believe that the risk of PASD may be high for placenta previa patients with endometriosis who conceived using frozen ET.

#### 4.3.5. Proposed Antenatal Management of PASD and Endometriosis

A prenatal diagnosis of PASD contributes to the reduction of hemorrhagic complications and improvements in the surgical outcomes of PASD, possibly due to the comprehensive multidisciplinary intraoperative treatment, including planned cesarean hysterectomy, transfusion preparation, and treatment administered by skilled surgeons, available for these women [22,109,110,111]. Previous studies suggested that the prenatal diagnosis rate of PASD are low for women with PASD who conceived using ART [19,50]. Therefore, clinicians should focus on the presence of PASD in women with endometriosis using ART who have risk factors for PASD (e.g., placenta previa, prior uterine surgery, advanced maternal age, etc.).

Our previous retrospective study reported that MRI can predict posterior uterine adhesions in patients with placenta previa. During this study, 96 patients with placenta previa were included; 21 of those patients had posterior uterine adhesions, probably because of endometriosis [22]. Nevertheless, we focused on the angle of the uterine cervix during the study and found that women with a retroverted cervix were more likely to have uterine posterior adhesions than women with an anteverted cervix (60% *versus* 8.5%; *p* < 0.01) [22]. Although the cost of MRI is high in developed countries, it has the potential to detect the presence of endometriosis in women with placenta previa [112].

#### 4.3.6. Proposed Intraoperative Management of PASD and Endometriosis

Cesarean hysterectomy for women with PASD is widely performed; however, its surgical outcomes for women with endometriosis have not been thoroughly determined. No studies have proposed an intraoperative surgical technique for PASD patients with endometriosis; therefore, morbidity during cesarean hysterectomy may be high. 

Cesarean hysterectomy without placental removal is the standard treatment for women with PASD. Nevertheless, “conservative management” is an option that aims to decrease maternal morbidity for women with PASD [1,113,114]. A retrospective, multicenter study of 167 women with PASD who received conservative management showed that only 42% received transfusions and 78% avoided hysterectomy [113]. Although 6% of women had severe maternal morbidity, such as delayed infection and hemorrhage, these results were considered important to the treatment of PASD. 

Moreover, a recent multicenter, prospective study (the PACCRETA prospective study) showed that conservative management of PASD is associated with a lower risk of transfusion (>4 units of red blood cells) within 6 months compared to cesarean hysterectomy [115]. Because the surgical morbidity of PASD patients with endometriosis may be high, we believe that this approach has the potential to improve surgical outcomes.

## 5. Conclusions and Implications

### 5.1. Implications for Practice

Endometriosis may correlate with unfavorable surgical outcomes, such as massive hemorrhage, bladder injury, and rectal injury, in cesarean hysterectomy. Moreover, an increased incidence of PASD was found in women with endometriosis compared with those without endometriosis. Thus, future research focusing on the surgical morbidity of pregnant women with PASD and endometriosis is necessary. Furthermore, it remains essential to identify intraoperative treatments or surgical techniques that could improve the surgical morbidity of pregnant women with PASD complicated with endometriosis. 

### 5.2. Implications for Clinical Research

Although the diagnosis of endometriosis during pregnancy is essential, there is a paucity of studies evaluating the diagnostic techniques applied in pregnant women. Since MRI is costly and not suitable for screening, future studies evaluating the usefulness of diagnosing endometriosis during pregnancy applying transvaginal ultrasonography are warranted.

## Figures and Tables

**Figure 1 biomedicines-10-00390-f001:**
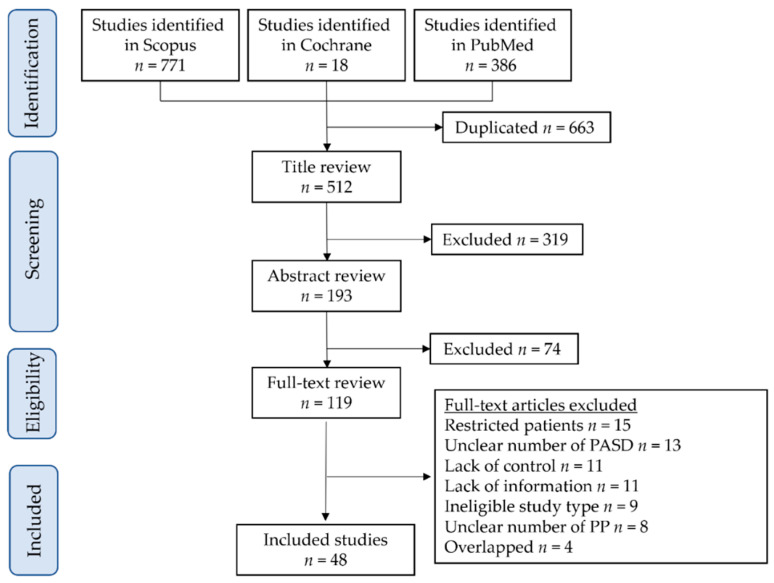
Study selection scheme of the systematic search of articles. Abbreviations: PP, placenta previa, PASD, placenta accreta spectrum disorder.

**Figure 2 biomedicines-10-00390-f002:**
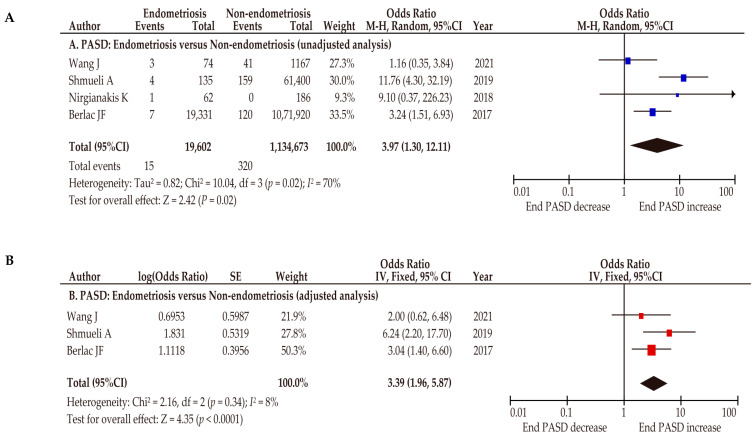
Effect of endometriosis on the prevalence of placenta accreta spectrum disorder. The pooled odds ratios (ORs) for the (**A**) unadjusted analysis of PASD and (**B**) adjusted analysis of PASD for women with endometriosis and women without endometriosis. Forest plots were ordered within the stratum by relative weight (%) and the publication year of the study. Substantial heterogeneity was observed during the unadjusted analysis (**A**: *I*^2^ = 70%), and low heterogeneity was observed during the adjusted analysis (**B**: *I*^2^ = 8%). Some values might be slightly different from the original values because calculations were performed using RevMan version 5.4.1. Abbreviations: PASD, placenta accreta spectrum disorder; End, endometriosis; OR, odds ratio; CI, confidence interval; SE, standard error.

**Figure 3 biomedicines-10-00390-f003:**
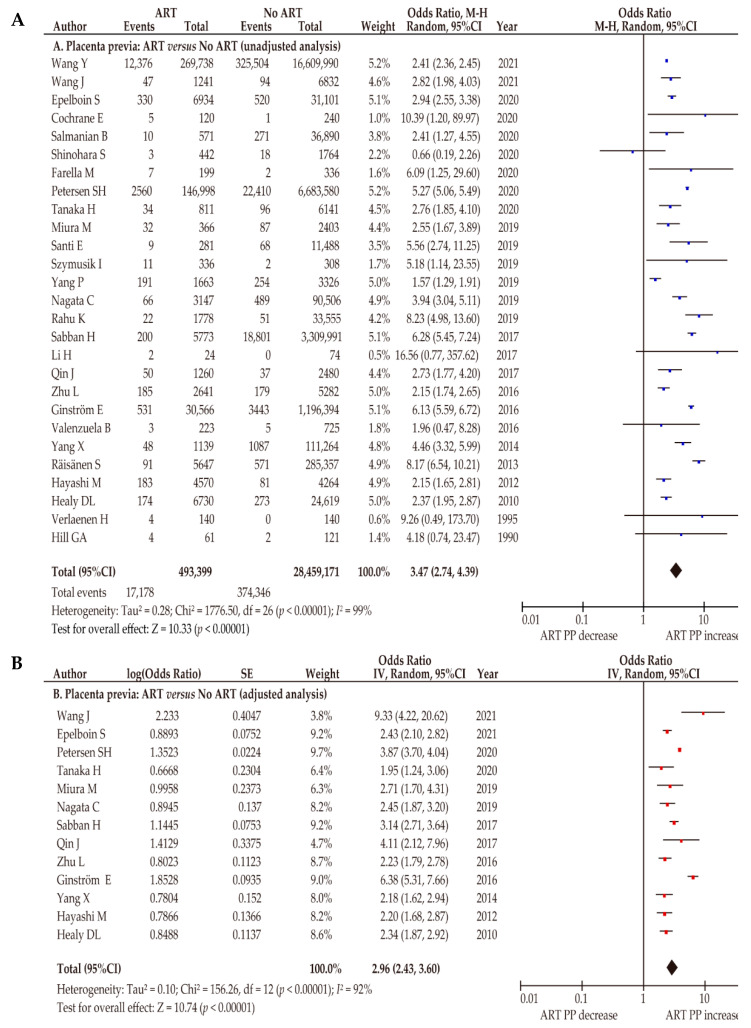
Effect of assisted reproductive technology on the prevalence of placenta previa. The pooled odds ratios (ORs) for the (**A**) unadjusted analysis of placenta accreta spectrum disorder (PASD) and (**B**) adjusted analysis of placenta previa for women who conceived using ART and women who conceived without ART. Forest plots were ordered within the stratum by relative weight (%) and the publication year of the study. Considerable heterogeneity was observed during the unadjusted analysis (**A**: *I*^2^ = 99%) and during the adjusted analysis (**B**: *I*^2^ = 92%). Some values might be slightly different from the original values, because the calculations were performed using RevMan ver. 5.4.1. Abbreviations: ART, assisted reproductive technology; PP, placenta previa; OR, odds ratio; CI, confidence interval; SE, standard error.

**Figure 4 biomedicines-10-00390-f004:**
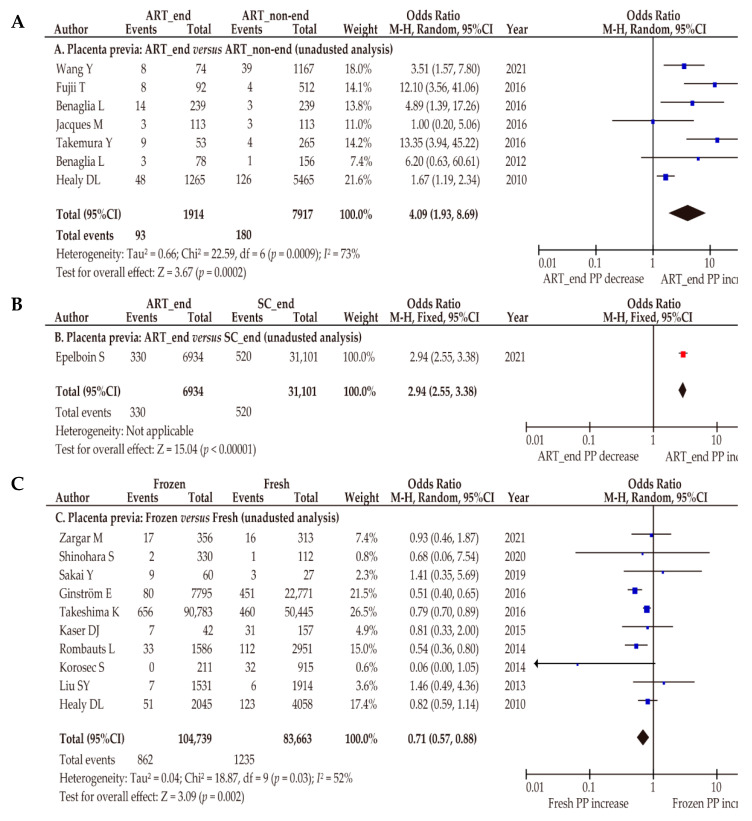
Effect of assisted reproductive technology on the prevalence of placenta previa (sensitivity analysis). All analyses were performed using an unadjusted analysis. All analyses compared the rates of placenta previa of the examination and control groups. The pooled odds ratios (ORs) for the comparisons were determined for the following groups: (**A**) ART with endometriosis *versus* ART without endometriosis; (**B**) women with endometriosis who conceived using ART *versus* women with endometriosis who conceived spontaneously; (**C**) frozen embryo transfer (ET) *versus* fresh ET; and (**D**) hormone replacement cycle (HRC) *versus* normal cycle (NC). Forest plots were ordered within the stratum by relative weight (%) and the publication year of the study. Substantial heterogeneity was observed during analysis (**A**) (*I*^2^ = 73%), moderate heterogeneity was observed during analysis (**C**) (*I*^2^ = 52%), and considerable heterogeneity was detected during analysis (**C**) (*I*^2^ = 54%). Some values might be slightly different from the original values, because calculations were performed using RevMan ver. 5.4.1. Abbreviations: ART, assisted reproductive technology; OR, odds ratio; CI, confidence interval; SE, standard error; PP, placenta previa; ART_end, endometriosis who conceived using ART; SC_end, endometriosis who conceived spontaneously; Frozen, frozen embryo transfer; Fresh, fresh embryo transfer.

**Figure 5 biomedicines-10-00390-f005:**
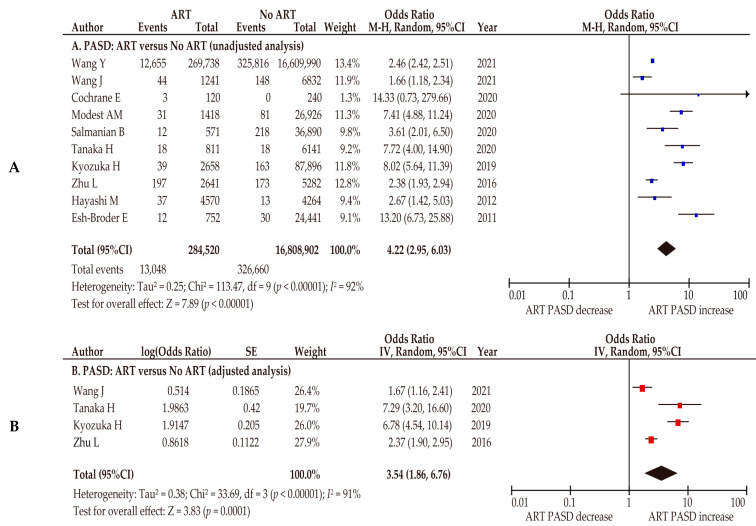
Effect of assisted reproductive technology on the incidence of placenta accreta spectrum. The pooled odds ratios (ORs) for the (**A**) unadjusted analysis of PASD and (**B**) adjusted analysis of PASD for women who conceived using ART and women who conceived without ART. Forest plots were ordered within the stratum by relative weight (%) and the publication year of the study. Considerable heterogeneity was observed during the unadjusted analysis (**A**: *I*^2^ = 92%) and the adjusted analysis (**B**: *I*^2^ = 91%). Some values might be slightly different from the original values, as calculations were performed using RevMan ver. 5.4.1. Abbreviations: ART, assisted reproductive technology; OR, odds ratio; CI, confidence interval; SE, standard error; PASD, placenta accreta spectrum disorder.

**Figure 6 biomedicines-10-00390-f006:**
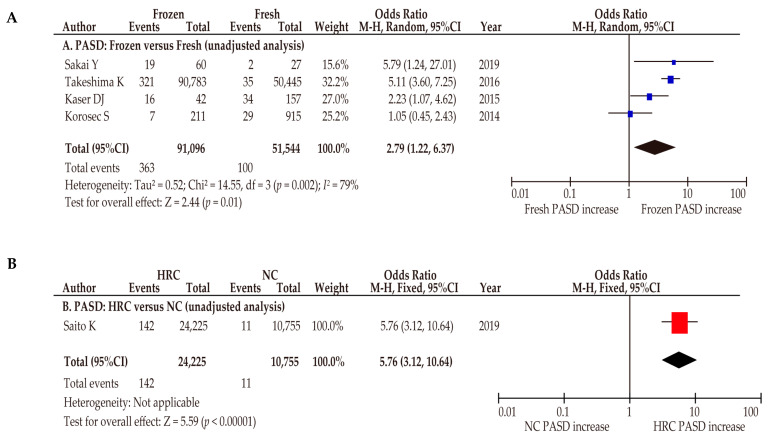
Effect of assisted reproductive technology on the incidence of placenta accreta spectrum (sensitivity analysis). The pooled odds ratios (ORs) for the unadjusted analysis of (**A**) comparisons between frozen embryo transfer (ET) and fresh ET and of (**B**) comparisons between the hormone replacement cycle (HRC) and normal cycle (NC) in one study. Forest plots were ordered within the stratum by relative weight (%) and the publication year of the study. Considerable heterogeneity was observed during both unadjusted analyses (**A**: *I*^2^ = 79%). Some values might be slightly different from the original values, as calculations were performed using RevMan ver. 5.4.1. Abbreviations: ART, assisted reproductive technology; OR, odds ratio; CI, confidence interval; SE, standard error; PASD, placenta accreta spectrum.

**Figure 7 biomedicines-10-00390-f007:**
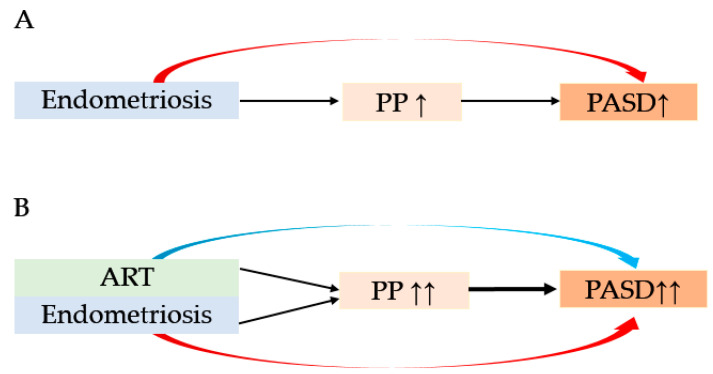
Our hypothesis of the possible mechanism of the increased rate of placenta accreta spectrum disorder. The red arrow indicates the effect of endometriosis on the increased rate of PASD. The light blue arrow indicates the effect of assisted reproductive technology (ART) on the increased rate of PASD. (**A**) For women with endometriosis who conceived without ART, endometriosis itself may be associated with an increased rate of PASD. Because endometriosis is associated with an increased rate of placenta previa, this may indirectly contribute to a higher rate of PASD. (**B**) For women with endometriosis who conceived using ART, ART may have an additive effect that increased the rate of placenta previa and PASD. As a result, we believe that women with endometriosis who conceived using ART are at a high risk for PASD. Abbreviations: PP, placenta previa; PASD, placenta accreta spectrum disorder, ART, assisted reproductive technology.

**Table 1 biomedicines-10-00390-t001:** The prevalence and trends of PASD.

Author	Year	Location	Type	PASD No.	Total No.	Rate	Trend	Definition
Ornaghi S [89]	2021	ITA	Nationwide	384	458,995	0.84%	--	Clinical
Matsuzaki S [17]	2021	USA	Nationwide	8030	2,727,477	0.29% ^†^	Increase	ICD-10
Miller HE [90]	2021	USA	Nationwide	1126	918,452	0.12%	--	ICD-10
Baldwin HJ [91]	2020	AUS	Nationwide	3300 *	1,306,308 *	0.25%	Increase	ICD-10
Kyozuka H [56]	2019	JPN	Nationwide	202	90,554	0.22%	--	Clinical
Jauniaux E [92]	2019	GBR	SR	20 studies	--	0.05–0.42%	--	--

^†^ Restricted to women with cesarean deliveries, * Number of births. Abbreviations: Type, type of study; PASD No., number of women with placenta accreta spectrum disorder; Total No., number of total cases; Rate, rate of placenta accreta spectrum disorders; Definition, definition of placenta accreta spectrum disorder: Nationwide, Nationwide study; SR, systematic review; Clinical, clinical diagnosis; ICD-10, International Classification of Diseases 10th Revision; ITA, Italy; USA, United States of America; AUS, Australia; JPN, Japan; GBR, United Kingdom.

**Table 2 biomedicines-10-00390-t002:** Effect of endometriosis on the prevalence of placenta accreta spectrum.

Author	Year	ART Rate	cOR	aOR (95%CI)	Def_End	Def_PASD
Wang J [47]	2021	100%	1.16 (0.35–3.84)	2.00 (0.62–6.48)	--	Clinical and histo
Shmueli A [63]	2019	26 (19.3%)	11.76 (4.30–32.19)	6.24 (2.2–17.7)	Clinical and histo	--
Nirgianakis K [64]	2018	22 (35.5%)	9.10 (0.37–226.23)	--	Histo	--
Berlac JF [27]	2017	3619 (19%)	3.24 (1.51–6.93)	3.04 (1.40–6.60)	ICD-10	--

Some values listed might be slightly different from the original values because the calculation was performed using RevMan version 5.4.1 or they were estimated by the authors. Abbreviations: OR, odds ratio; cOR, crude odds ratio; aOR, adjusted odds ratio; --, not applicable; ART, assisted reproductive technology; CI, confidence interval; Def_End, the definition of endometriosis; Def_PASD, the definition of placenta accreta spectrum disorder; Clinical, clinical diagnosis; Histo, histologically confirmed endometriosis; ICD-10, International Classification of Diseases 10th Revision.

**Table 3 biomedicines-10-00390-t003:** Rates of placenta previa (ART *versus* no ART).

Author	Year	Exp	Control	cOR (95%CI)	aOR (95%CI)
Wang Y [43]	2021	ART	No ART	2.41 (2.36–2.45)	--
Wang J [47]	2021	ART	No ART	2.82 (1.98–4.03)	9.33 (4.22–20.62)
Epelboin S [48] ^a^	2021	ART_end	SC_end	2.94 (2.55–3.38)	2.43 (2.10–2.82)
Cochrane E [49]	2020	ART	SC	10.39 (1.20–89.97)	--
Salmanian B [51]	2020	ART	No ART	2.41 (1.27–4.55)	--
Shinohara S [52]	2020	ART	SC	0.66 (0.19–2.26)	--
Farella M [53] ^a^	2020	ART_end	SC_end	6.09 (1.25–29.60)	--
Petersen SH [54]	2020	ART	SC	5.27 (5.06–5.49)	3.87 (3.70–4.04)
Tanaka H [55]	2020	ART	No ART	2.76 (1.85–4.10)	1.95 (1.24–3.06)
Miura M [26]	2019	ART	No ART	2.55 (1.67–3.89)	2.71 (1.70–4.31)
Santi E [57]	2019	ART	No ART	5.56 (2.74–11.25)	--
Szymusik I [60]	2019	ART	SC	5.18 (1.14–23.55)	--
Yang P [61]	2019	ART	No ART	1.57 (1.29–1.91)	--
Nagata C [88]	2019	ART	SC	3.94 (3.04–5.11)	2.45 (1.87–3.20) ^b^
Rahu K [62]	2019	ART	SC	8.23 (4.98–13.60)	--
Sabban H [65]	2017	ART	No ART	6.28 (5.45–7.24)	3.14(2.71–3.64)
Li H [66]	2017	ART_end	No ART_end	16.56 (0.77–357.62)	--
Qin J [67]	2017	ART	No ART	2.73 (1.77–4.20)	4.11 (2.12–7.96)
Zhu L [69]	2016	ART	SC	2.15 (1.74–2.65)	2.23 (1.79–2.78)
Ginström E [72]	2016	ART	SC	6.13 (5.59–6.72)	6.38 (5.31–7.66) ^c^
Valenzuela B [73]	2016	ART	No ART	1.96 (0.47–8.28)	--
Yang X [78]	2014	ART	SC	4.46 (3.32–5.99)	2.18 (1.62–2.94)
Räisänen S [79]	2013	ART	No ART	8.17 (6.54–10.21)	--
Hayashi M [82]	2012	ART	SC	2.15 (1.65–2.81)	2.2 (1.68–2.87)
Healy DL [85]	2010	ART	No ART	2.37 (1.95–2.87)	2.34 (1.87–2.92)
Verlaenen H [86]	1995	ART	No ART	9.26 (0.49–173.70)	--
Hill GA [87]	1990	ART	No ART	4.18 (0.74–23.47)	--

^a^ All cases were women with endometriosis. ^b^ Data (OR for ART and intracytoplasmic sperm injection) were combined by the author. ^c^: women with cleavage stage were excluded. Some values listed might be slightly different from the original values because the calculation was performed using RevMan version 5.4.1 or they were estimated by the authors. Abbreviations: OR, odds ratio; cOR, crude odds ratio; aOR, adjusted odds ratio; Exp, experimental group; Control, control group; ART, assisted reproductive technology; SC, spontaneous conception; CI, confidence interval; --, not applicable; ART_end; women with endometriosis conceived by ART; SC_end; spontaneous conception with endometriosis.

**Table 4 biomedicines-10-00390-t004:** Rates of placenta previa for women who conceived using ART (sensitivity analysis).

Author	Year	No. Exp	No. Control	Exp End (%)	cOR	aOR
**Endometriosis *versus* Non-endometriosis (women with ART)**						
Wang J [47]	2021	74	1167	74/1167 (6.0%)	3.51 (1.57–7.80)	--
Fujii T [68]	2016	92	512	92/604 (15.2%)	12.10 (3.56–41.06)	15.41 (4.40–61.7)
Benaglia L [70]	2016	239	239	--	4.89 (1.39–17.26)	--
Jacques M [71]	2016	113	113	160/2315 (6.9%)	1.00 (0.20–5.06)	--
Takemura Y [81]	2013	53	265	53/318 (16.7%)	6.20 (0.63–60.61)	--
Benaglia L [83]	2012	78	156	--	13.35 (3.94–45.22)	--
Healy DL [85]	2010	1265	5465	1265/6730 (18.8%)	1.67 (1.19–2.34)	1.65 (1.18–2.32)
**ART *versus* SC (in women with endometriosis)**						
Epelboin S [48]	2021	6934	31,101	6934/38,035(18.2%)	2.94 (2.55–3.38)	--
**ART with endometriosis *versus* SC without endometriosis**						
Epelboin S [48]	2021	6934	4,083,732	--	8.82 (7.89–9.86)	--
**Frozen ET *versus* Fresh ET**						
Zargar M [44]	2021	356	313	--	0.93 (0.46–1.87)	--
Shinohara S [52]	2020	330	112	--	0.68 (0.06–7.54)	--
Sakai Y [58]	2019	60	27 ^b^	--	1.41 (0.35–5.69)	--
Ginström E [72]	2016	7795	22,771	--	0.51 (0.40–0.65)	--
Takeshima K [74]	2016	90,783	50,445	--	0.79 (0.70–0.89)	--
Kaser DJ [75]	2015	42	157	--	0.81 (0.33–2.00)	
Rombauts L [76]	2014	1586	2951	99/1586 (6.2%)	0.54 (0.36–0.80)	--
Korosec S [77]	2014	211	915	43/211 (20.4%)	0.06 (0.00–1.05)	--
Liu SY [80]	2013	1531 ^a^	1914 ^a^	--	1.46 (0.49–4.36)	
Healy DL [85]	2010	2045	4058	--	0.82 (0.59–1.14)	1.37 (0.96–1.95) ^c^
**HRC *versus* NC (Frozen ET)**						
Hu KL [45]	2021	2561	3790	122 (4.8%)		0.58 (0.27–1.24) ^c^
Asserhøj LL [46]	2021	357	731	16 (5.0%)		1.10 (0.46–2.62)
Saito K [59]	2019	24,225	10,755	--	0.80 (0.59–1.08)	1.00 (0.64–1.56)
Rombauts L [76]	2014	355	1231	--	2.02 (0.98–4.14)	--

^a^ Day 3 embryo. ^b^ Two cases involved frozen embryo transfer. ^c^ Calculated by the author. Some values listed might be slightly different from the original values because the calculation was performed using Revman ver. 5.4.1 or they were estimated by the authors. Abbreviations: Exp, Experimental; End, endometriosis; Exp End, the rate of endometriosis in the experimental group; ART, assisted reproductive technology; --, not applicable; Frozen, frozen embryo transfer; Fresh, fresh embryo transfer; ET, embryo transfer; No., number; OR, odds rate; cOR, crude odds ratio; aOR, adjusted odds ratio; No. Exp; number of women included in the experimental group; No. Control, number of women included in the control group; ART_non-end; women without endometriosis conceived by ART; SC_end, spontaneous conception with endometriosis; SC_non-end, spontaneous conception without endometriosis; HRC, hormone replacement cycle; NC, natural cycle.

**Table 5 biomedicines-10-00390-t005:** Effect of ART on PASD rates.

Author	Year	Exp	Control	No. Exp	No. Control	cOR	aOR
**ART *versus* No ART and SC**							
Wang Y [43]	2021	ART	No ART	269,738	16,609,990	2.46 (2.42–2.51)	--
Wang J [47]	2021	ART	No ART	1241	6832	1.66 (1.18–2.34)	1.67 (1.16–2.41) ^a^
Cochrane E [49]	2020	ART	SC	120	240	14.33 (0.73–279.66)	--
Modest AM [50]	2020	ART	No ART	1418	26,926	7.41 (4.88–11.24)	--
Salmanian B [51]	2020	ART	No ART	571	36,890	3.61 (2.01–6.50)	--
Tanaka H [55]	2020	ART	No ART	811	6141	7.72 (4.00–14.90)	7.35 (3.20–16.6)
Kyozuka H [56]	2019	ART	No ART	2658	87,896	8.02 (5.64–11.39)	6.78 (4.54–10.14)
Zhu L [69]	2016	ART	SC	2641	5282	2.38 (1.93–2.94)	2.37 (1.90–2.95)
Hayashi M [82]	2012	ART	SC	4570	4264	2.67 (1.42–5.03)	--
Esh-Broder E [84]	2011	ART	SC	752	24,441	13.20 (6.73–25.88)	--
**Frozen ET *versus* Fresh ET**							
Sakai Y [58]	2019	Frozen	Fresh	60	27 ^b^	5.79 (1.24–27.01)	--
Takeshima K [74]	2016	Frozen	Fresh	90,783	50,445	5.11 (3.60–7.25)	--
Kaser DJ [75]	2015	Frozen	Fresh	42	157	2.23 (1.07–4.62)	3.20 (1.14–9.02)
Korosec S [77]	2014	Frozen	Fresh	211	915	1.05 (0.45–2.43)	--
**HRC *versus* NC (Frozen ET)**							
Saito K [59]	2019	HRC	NC	24,225	10,755	5.76 (3.12–10.64)	6.91 (2.87–16.66)

Updated and modified from Sci Rep. 2021;11:920 Matsuzaki S et al. Antenatal diagnosis of placenta accreta spectrum after in vitro fertilization –embryo transfer: a systematic review and meta-analysis/Appendix A, Meta-table of the included studies [15]. ^a^ Calculated by the author. ^b^ Two cases involved frozen embryo transfer. Some values listed might be slightly different from the original values because the calculation was performed using Revman ver. 5.4.1. Some values listed might be slightly different from the original values, because they were estimated by the authors. Abbreviations: Exp, Experimental; End, endometriosis; PASD, placenta accreta spectrum; ART, assisted reproductive technology; SC, spontaneous conception; --, not applicable; Frozen, frozen embryo transfer; Fresh, fresh embryo transfer; ET, embryo transfer; No., number; No. Exp, number of women in the experimental group; No. Control, number of women in the control group; HRC, hormone replacement cycle; NC, natural cycle; OR, odds ratio; cOR, crude odds ratio; aOR, adjusted odds ratio.

## Data Availability

All the studies used in this study are published in the literature.

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
