# Peer review of "Placenta Accreta Spectrum Disorder Complicated with Endometriosis: Systematic Review and Meta-Analysis"

_biomedicines, 2022, doi:10.3390/biomedicines10020390_

Round 1

Reviewer 1 Report

Congratulations on this very interesting and well written study. I enjoyed reading your review and I'm convinced that it is very relevant to colleagues working in the field of obstetrics and ART. I have only one important remark: in my opinion figure 1 does not really contribute to your work and is rather redundant for the reader, so I would recommend omitting it. 

Author Response

We would like to thank the Editor and the Reviewers for the helpful comments. The following are our point-by-point responses to the comments and explanations regarding the revisions made to the manuscript. The line numbers of the revised text are indicated. The revisions made in the manuscript are indicated using the “track changes” function of Microsoft Word.

Reviewer #1

Congratulations on this very interesting and well written study. I enjoyed reading your review and I'm convinced that it is very relevant to colleagues working in the field of obstetrics and ART.

Reply:

We appreciate these useful comments from the Reviewer. We have revised the manuscript carefully according to the Reviewer’s comments. We believe the revised manuscript now addresses the Reviewer’s concerns.

Reviewer #1, Comment 1

I have only one important remark: in my opinion figure 1 does not really contribute to your work and is rather redundant for the reader, so I would recommend omitting it.

Reply: Figure 1

Thank you for your helpful comments. As the Reviewer has pointed out, we have removed Figure 1 from the main text.

Reviewer 2 Report

This systematic review deals with the relationship between placenta accreta spectrum disorder (PASD) and endometriosis. The relationships among pregnancy, assisted reproductive technology (ART), placenta previa, ART-conceived pregnancy and PASD were also determined. A systematic literature review was conducted using multiple computerized databases. Forty-eight studies met the inclusion criteria. Endometriosis was associated with an increased prevalence of PASD. Women who used ART were more likely to have placenta previa and PASD than those who did not use ART. Women with placenta previa complicated with endometriosis who conceived using frozen ET may be a high risk for PASD. The methodology (systematic literature review, selection of eligibility criteria, information sources, search study selection, data extraction, analysis of outcome measures and assessment of bias risk, meta-analysis and statistical analysis) of the analysis is of a high quality. The description of the results is somewhat overcomplicated, could be simplified (e.g., Table 4).

Author Response

We would like to thank the Editor and the Reviewers for the helpful comments. The following are our point-by-point responses to the comments and explanations regarding the revisions made to the manuscript. The line numbers of the revised text are indicated. The revisions made in the manuscript are indicated using the “track changes” function of Microsoft Word.

Reviewer #2

This systematic review deals with the relationship between placenta accreta spectrum disorder (PASD) and endometriosis. The relationships among pregnancy, assisted reproductive technology (ART), placenta previa, ART-conceived pregnancy and PASD were also determined. A systematic literature review was conducted using multiple computerized databases. Forty-eight studies met the inclusion criteria. Endometriosis was associated with an increased prevalence of PASD. Women who used ART were more likely to have placenta previa and PASD than those who did not use ART. Women with placenta previa complicated with endometriosis who conceived using frozen ET may be a high risk for PASD.

The methodology (systematic literature review, selection of eligibility criteria, information sources, search study selection, data extraction, analysis of outcome measures and assessment of bias risk, meta-analysis and statistical analysis) of the analysis is of a high quality.

Reply:

Thank you for your positive comments. We have revised the manuscript according to your valuable comments. Please refer to our reply for each comment.

Reviewer #2, Comment 1

The description of the results is somewhat overcomplicated, could be simplified (e.g., Table 4).

Reply: Table 4

We appreciate these valuable comments from the Reviewer. According to the Reviewer’s comment, we have revised Table 4.